# Evaluating data quality and reference instrument robustness: insights of 12 years DI magnetometer comparisons in the Geomagnetic Network of China

Yufei He[1], Xudong Zhao[1], Suqin Zhang[1], Qi Li[1] and Fuxi Yang[2]

[1]Institute of Geophysics, China Earthquake Administration, Beijing, 100081, China
[2]Earthquake Bureau of Xinjiang Province, Urumqi, 830011, China

*Correspondence to*: Xudong Zhao (zxd9801@163.com)

**Abstract.** A statistical analysis was conducted on 12 years of geomagnetic instrument comparison data from the Chinese Geomagnetic Network (GNC) cover 15 years from 2010 to 2024. The study reveals that when the probability density of instrument differences accumulates to 90%, the corresponding instrument difference are 0.21 ′ (D component) and 0.11′ (I component), which can serve as evaluation criteria at the network level. By integrating multi source uncertainty decomposition (including instrument errors, operator dependent errors, and pillar correction errors) and weighted ensemble analysis, systematic differences between reference fluxgate theodolites and test instruments were quantified. Results demonstrate that reference instruments exhibit high stability and reliability, with mean differences of -0.004 ′ (D) and 0.022 ′ (I), both within the 95% confidence interval, and no long term drift was observed. Operator dependent errors were successfully isolated, with 0.13 ′ (D) and 0.06 ′ (I), consistent with observed experimental findings, confirming that operator dependent errors constitute the primary contributor to instrument differences. Notably, operator dependent errors in D are significantly higher than I due to the complexity of azimuth alignment and levelling. These findings highlight the critical role of instrument comparisons in effectively monitoring equipment performance and assessing observational quality across observatoris, while validating the feasibility of standardized instrument evaluation methods and the long term stability of reference instruments. Future efforts should integrate sensors and automation technologies to minimize human errors, thereby providing a higher quality data foundation for geophysical studies.

## 1 Intruduction

In geomagnetic observatories, variometers are employed to record continuous variations of the geomagnetic field. These variations are subsequently converted into absolute geomagnetic field values through the addition of baseline values derived from absolute measurements (Jankowski and Sucksdorff, 1996). This calibration process renders absolute measurements critical for ensuring the quality of continuous absolute geomagnetic data. However, the difference of absolute instruments

between different observatories make systematic instrument comparisons as an essential component of modern geomagnetic observation systems.

Contemporary absolute measurements primarily utilize two high precision instruments: (i) fluxgate theodolites (designated as Declination Inclination Magnetometers, DIMs) for measuring declination (D) and inclination (I), and (ii) proton magnetometers for total field intensity (F) determinations. While technological advancements have reduced the required frequency of instrument comparisons, such calibrations remain indispensable for maintaining high quality geomagnetic datasets (Zhang et al., 2024). To standardize global geomagnetic observations, the IAGA Division V Working Group V-OBS has successfully organized over twenty biennial international instrument comparison sessions to date (Loubser et al., 2002; Masami et al., 2004, Reda et al., 2007; Love et al.,2009; He et al., 2011; Hejda et al.,2013; and so on). In China, the Geomagnetic Network Center (GNC) integrates these comparisons into its quality control framework, serving dual roles as both data hub and quality assurance authority for national observatories (Li et al., 2012; Zhang et al., 2016).

Since the digital transformation of Chinese geomagnetic observatories, the GNC has implemented successive generations of DIMs, including Hungarian MINGEO-DIM, British MAG 01-DIM, Chinese CTM-DIM, Chinese GEO-DIM, Chinese TDJ2E-NM-DIM, etc. It should be noted that the manual operation inherent to these DIM systems introduces two critical uncertainty sources: i) inter instrument systematic biases, and ii) operator dependent operational variances. To mitigate these effects and unify observational standards across the geomagnetic network, the GNC has conducted multi annual comparative measurements and accumulated corresponding datasets since the digital transformation of GNC (He et al., 2019b). An instrument demonstrating high measurement accuracy and operational stability is typically designated as the national reference standard for GNC. Through systematic comparative measurements, each observatory's instruments are calibrated against this reference standard to quantify instrumental differences, thereby achieving nationwide standardization of geomagnetic absolute observations. As the reference instrument, it requires not only meticulous routine maintenance but also periodic metrological verification to ensure sustained measurement precision.

This study investigates the robustness of the reference instrument through historical comparative measurement data. The methodology initiates with an overview of the comparative measurement protocol and multiyear calibration results. Subsequently, uncertainty propagation analysis, a widely used uncertainty quantification method in experimental science, is applied to this dataset. This technique determines the uncertainty of derived results by propagating the uncertainties of input measurements through computational processes. So the uncertainty for each comparative session can be yielded. Concurrently, systematic differences of the reference instrument relative to iterative measurements are quantified to evaluate its long term stability.

## 2 Measurement and comparative methodology

### 2.1 Measurement principles

The geomagnetic field, being a vector quantity, requires precise determination of both magnitude and directional components. The DIM, comprising a theodolite integrated with a fluxgate sensor, serves as the standard instrument for determining geomagnetic field direction (declination D and inclination I). The fluxgate sensor, mounted parallel to the theodolite's optical axis, operates on the null detection principle: it generates zero output (assuming zero offset) when aligned perpendicular to the geomagnetic field vector. Directional determination is achieved by identifying sensor null positions, with angular coordinates recorded via theodolite circle readings. The angular between two directions can be determined by computing the difference between their respective readings on the instrument's horizontal circle. This is the fundamental principle of theodolite angle measurement.

### 2.2 Declination and inclination measurement

Geomagnetic declination and inclination measurements are performed within the horizontal and magnetic meridional of the theodolite respectively. The declination determination involves two sequential operations: establishing true north orientation and identifying the geomagnetic meridian direction. The true north orientation is calibrated by aligning the telescope's optical axis with a predefined azimuth marker. In order to eliminate errors associated with the optical misalignment of the theodolite, two observations are required to find the true north direction, one with sensor up and the other with sensor down. Finally, the direction of the azimuth marker can be determined through two readings and recorded as M (Fig.1). As the azimuth value A of the marker known, the true north position can be calculated. Subsequently, the geomagnetic meridian direction is identified by searching the fluxgate null position in the horizontal plane (with the vertical circle maintained at 90 °or 270 °) and recording horizontal reading (D'). The geomagnetic declination D is then derived from the differential angular measurement following the formula:

$$D = D' - M + A.  \tag{1}$$

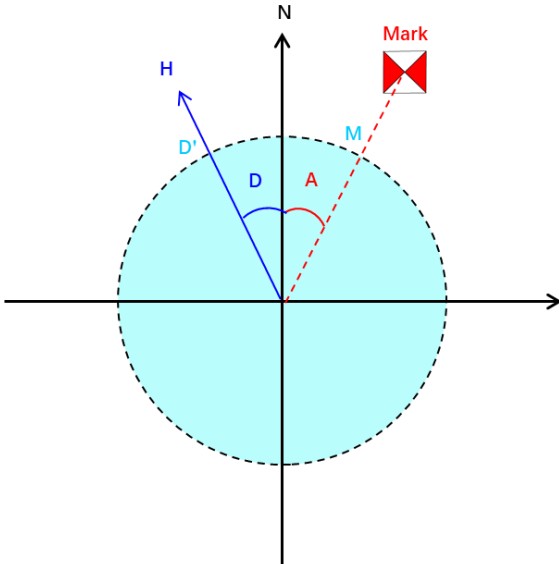

**Figure 1: Measurement principle of the declination.**

Inclination measurements follows analogous procedures and is carried out in the magnetic meridional plane derived from the previous declination measurements, while also within the vertical reference provided by the gravity field through the theodolite suspension system.

## 2.3 Error mitigation strategy

The theodolites are high-precision instruments, but they inevitably contain certain errors, such as misalignment errors between the mechanical axis of theodolite, the optical axis of the telescope, and the magnetic axis of the fluxgate sensor; collimation errors; non-orthogonality errors of the horizontal and vertical axes; uneven graduation errors of the reading circle; index errors; and errors caused by non-zero electronic offsets, which prevent accurate determination of magnetic declination and inclination from a single reading of the horizontal/vertical circle (Lauridsen, 1985; Newitt et al., 1996; Csontos and Sugar, 2024). However, in theory, most of these errors can be eliminated through the four position measurement process, and some of them (two misalignment errors between the fluxgate sensor axis and the optical axis of the telescope in the horizontal/vertical planes, and the offset error of the fluxgate sensor) can be calculated from the measurement results (Bitterly et al., 1984). Nevertheless, errors cannot be completely eliminated and will still exist, which is the main reason for the differences between different instruments and the source of uncertainty in measurement results. The instrument differences defined in this paper are the comprehensive differences of the entire instrument system, representing the differences between results obtained by the instruments after four measurement processes, under the assumption of no personnel operation error. Consequently, the text does not explore the impacts of various internal errors on the measurement results. Consequently, the impact of various internal errors on measurement results is not separately explored in this article, but their combined effects are considered as the overall internal error of the theodolite.

The measurement procedure follows the guide published by IAGA (Newitt et al., 1996), and the specific description of the four positions observation can refer to the observation steps in Csontos and Sugar's (2024) paper. The declination measurement protocol is preceded and followed by sensor up and down azimuth marker readings and then involves four configurations: (i) telescope East/sensor up ($D_1$), (ii) telescope West/sensor down ($D_2$), (iii) telescope East/sensor down ($D_3$), and (iv) telescope West/sensor up ($D_4$). Four different position observations can eliminate errors associated with theodolite optics, sensor misalignment and electronics offset (Csontos and Sugar, 2024). Then final declination value is derived through arithmetic averaging:

$$D' = (D_1 + D_2 + D_3 + D_4)/4 + 90° \quad . \tag{2}$$

An analogous procedure governs inclination measurement, with positional configurations: (i) telescope North/sensor up ($I_1$), (ii) telescope South/sensor down ($I_2$), (iii) telescope North/sensor down ($I_3$), and (iv) telescope South/sensor up ($I_4$). The inclination is calculated as:

$$I = (I_1 + I_2 - I_3 - I_4)/4 + 90° \quad . \tag{3}$$

This methodology effectively compensates for fluxgate sensor optical axis misalignment (Deng et al., 2010).

Two distinct circle reading techniques are employed: the null method (exact zero point detection) and the offset method (near zero linear region utilization). As demonstrated by Xin (2003), Lu (2008), and Deng (2011), modern theodolites' high output linearity enables equivalent accuracy between methods, even with minor operator induced magnetic interference, making the offset method preferable for operational efficiency.

By using the geomagnetic declination (D), inclination (I), and the total magnetic intensity (F) measured by the proton magnetometer, all the absolute components of the Earth's magnetic field can be calculated. This will facilitate the subsequent baseline calculations of variometer for all components (such as east, north, and vertical directions).

## 2.4 Baseline comparison methodology

Ensuring high quality geomagnetic observations necessitates systematic verification of inter-instrument differences across observatories. The comparative analysis of absolute measurement instruments constitutes an essential quality assurance measure in geomagnetic monitoring networks. Practical constraints, including limited pillar availability, multiple DIMs, and operator skills proficiency, render synchronous multi instrument comparisons operationally challenging. Modern variometers exhibit high precision performance with quasi constant baseline characteristics under stable operating conditions, while underground observation rooms of geomagnetic observatories (far from cities or villages) can provide such operating conditions, including no influence of magnetic objects, low electromagnetic background noise, indoor annual temperature variation not exceeding 10 ℃, daily variation not exceeding 0.3 ℃, and so on. The Current comparison protocols therefore employ independent absolute measurements followed by baseline value cross validation. The stability and accuracy of baseline values during a single calibration day were investigated by Zhang (2011), whose study confirmed that baseline values remained stable throughout the calibration period (8:30 to 16:30 local time), with geomagnetic activity exhibiting no significant impact

on their accuracy. Consequently, direct comparison of baseline values is a valid approach for completing the analysis. The formula for calculating the baseline value of component $W$, as defined in the INTERMAGNET Reference Manual (St Louis, 2024), is presented below:

$$W_B(k) = W_o(i:j) - W_R(k),\tag{4}$$

Where $(i:j)$ is the time interval (typically minutes) for measurement, $(k)$ is the k-th time, the average time of interval $(i:j)$, $W_o(i:j)$ is the absolute field value for the time interval $(i:j)$, $W_R(k)$ is the variometer recorded value at time $k$, and $W_B(k)$ is the derived baseline value.

When absolute measurements are performed on different pillars, baseline correction to the reference pillar requires pillar differences. The generalized formulation for component W correction is:

$$\Delta U_{SO} = W_{BS} - W_{BO} + \Delta W_{SO},\tag{5}$$

where $s$ is the reference pillar designation, o is the non-reference observation pillar, $W_{BS}$ and $W_{BO}$ are respectively the baseline values from reference and observation pillars, $\Delta U_{SO}$ is the final instrument difference, $\Delta W_{SO}$ is the pillar difference, which represents the difference between the base pillar and the reference pillar. These pillar differences were measured before the observatory was put into operation and remeasured before each comparison.

There are two main ways to determine the pillar difference: the direct simultaneous measurements and indirect baseline values comparison. If two or more instruments are available for measurements, the direct method can be applied, and the pillar difference can be calculated by the following Eq. (6):

$$\Delta W_{SO} = [(W_{ps} + W_{qs}) - (W_{po} + W_{qo})]/2,\tag{6}$$

where $p$, $q$ denote the different instruments and $s$, $o$ represent the standard pillar and other observation pillars, respectively. $W_{ps}$ and $W_{qo}$ represent the baseline value of instrument p on standard pillar and the baseline value of instrument q on other pillars, respectively. $\Delta W_{SO}$ is the pillar difference between the standard pillar and other pillars.

If only one instrument is available, the indirect method can be used to calculated the pillar difference using the follow Eq. (7):

$$\Delta W_{SO} = W_{ps} - W_{po},\tag{7}$$

This methodology enables cross comparisons of fluxgate theodolite through pillar reference baseline correction. The obtained difference values $\Delta U_{SO}$ provide quantitative evaluation parameters for assessing absolute observation data quality across participating instruments.

**2.5 DIMs intercomparison results of GNC**

Based on the aforementioned methodology, the GNC has completed 12 comparisons of DIMs covering 15 years from 2010 to 2024 (no comparison was conducted in 2011, 2013, and 2021). The fluxgate theodolites involved in the comparison work of GNC include five types mentioned earlier: the MINGEO-DIM, the MAG-01H-DIM, the CTM-DIM, the GEO-DIM, and the TDJ2E-NM-DIM. These instruments are from 46 observatories, and the locations of all observatories are shown in Fig.2. Table

1 lists the codes and corresponding instrument types for all the observatories, as well as the number of times each instrument
participated in comparison and the number of operators (with non-repeated counts).

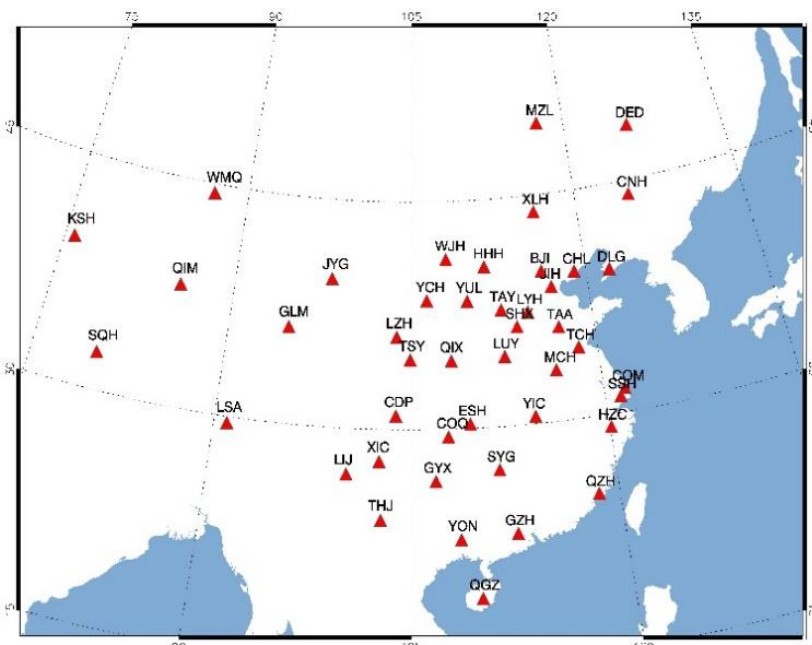

**Figure 2: The observatory locations of GNC.**

**Table 1: The list of instrument type and its observatory**

| No. | Observatory Code | Instrument | Operators Number (non-repeated) | Comparison Frequency | No. | Observatory Code | Instrument | Operators Number (non-repeated) | Comparison Frequency |
|---|---|---|---|---|---|---|---|---|---|
| 0 | GNC | Mingeo | 3 | 12 | 32 | QIM | Mag-01H | 2 | 9 |
| 1 | BJI | Mingeo | 3 | 8 | 33 | QGZ | GEO | 2 | 2 |
| 2 | CHL | Mag-01H | 3 | 10 | 34 | QGZ | Mingeo | 3 | 6 |
| 3 | CHL | CTM | 1 | 1 | 35 | QZH | Mingeo | 5 | 10 |
| 4 | CDP | Mingeo | 3 | 9 | 36 | QZH | CTM | 1 | 1 |
| 5 | CDP | Mag-01H | 1 | 1 | 37 | SYG | Mingeo | 3 | 10 |
| 6 | CDP | GEO | 2 | 2 | 38 | SSH | Mingeo | 2 | 9 |
| 7 | COM | Mag-01H | 4 | 5 | 39 | SSH | TDJ2E-NM | 1 | 2 |
| 8 | COM | TDJ2E-NM | 1 | 4 | 40 | SHX | Mag-01H | 3 | 10 |
| 9 | DLG | Mingeo | 4 | 10 | 41 | SQH | Mingeo | 2 | 3 |
| 10 | DED | Mingeo | 2 | 6 | 42 | TAY | Mingeo | 7 | 11 |
| 11 | ESH | Mingeo | 5 | 9 | 43 | TAA | GEO-DI | 2 | 4 |
| 12 | GLM | Mingeo | 3 | 8 | 44 | TAA | CTM | 1 | 1 |
| 13 | GYX | Mingeo | 3 | 5 | 45 | TSY | Mag-01H | 4 | 7 |

| 14 | HZC | CTM | 1 | 4 | 46 | THJ | Mingeo | 3 | 10 |
| 15 | HZC | Mag-01H | 1 | 1 | 47 | WJH | Mingeo | 4 | 10 |
| 16 | LYH | Mag-01H | 1 | 2 | 48 | WMQ | Mingeo | 3 | 10 |
| 17 | LYH | TDJ2E-NM | 3 | 5 | 49 | WHN | Mingeo | 5 | 6 |
| 18 | HHH | CTM | 2 | 2 | 50 | WHN | Mag-01H | 2 | 2 |
| 19 | HHH | TDJ2E-NM | 2 | 2 | 51 | WHN | CTM | 1 | 1 |
| 20 | JYG | Mingeo | 4 | 8 | 52 | XIC | CTM | 1 | 6 |
| 21 | JIH | Mingeo | 3 | 6 | 53 | XLH | Mag-01H | 3 | 6 |
| 22 | JIH | Mag-01H | 4 | 5 | 54 | COQ | Mag-01H | 5 | 7 |
| 23 | KSH | Mingeo | 4 | 9 | 55 | YCH | Mingeo | 2 | 8 |
| 24 | LSA | Mingeo | 5 | 9 | 56 | YIC | Mingeo | 2 | 4 |
| 25 | LZH | Mingeo | 5 | 11 | 57 | YON | Mingeo | 6 | 11 |
| 26 | LIJ | Mag-01H | 3 | 6 | 58 | YUL | CTM | 2 | 4 |
| 27 | LUY | Mingeo | 3 | 8 | 59 | YUL | TDJ2E-NM | 3 | 4 |
| 28 | TCH | Mingeo | 3 | 10 | 60 | CNH | Mingeo | 3 | 8 |
| 29 | MZL | Mingeo | 3 | 10 | 61 | GZH | Mingeo | 2 | 2 |
| 30 | MCH | Mag-01H | 3 | 9 | 62 | GZH | Mag-01H | 4 | 4 |
| 31 | QIX | Mingeo | 3 | 10 | 63 | GZH | TDJ2E-NM | 3 | 3 |


No.0 in Table 1 is designated as the standard reference instrument for GNC. The standard instrument consists of a Zeiss 010B theodolite and a fluxgate sensor, with accuracies better than 1 arc second and 0.1 nT, respectively. The measurements of the standard reference instrument are taken on the reference pillar considered the standard of the observatory. This pillar occupies the location with the minimal magnetic gradient within the absolute observation room and serves as the core reference point

for the entire geomagnetic observatory. All observed values of the station are ultimately normalized to this point.

Figures 3(a) and (b) respectively illustrate the instrumental differences in declination ($\Delta U_D$) and inclination ($\Delta U_I$) between observatory instruments and reference standards. Colored dots in the figures represent measurement results from different years, with vertical coordinates indicating instrumental differences, while the dot sizes indicate the standard deviations of

measurements, scaled according to the legend on the right. This graphical representation enables a comprehensive evaluation of observational data quality at both individual instrument and the network levels.

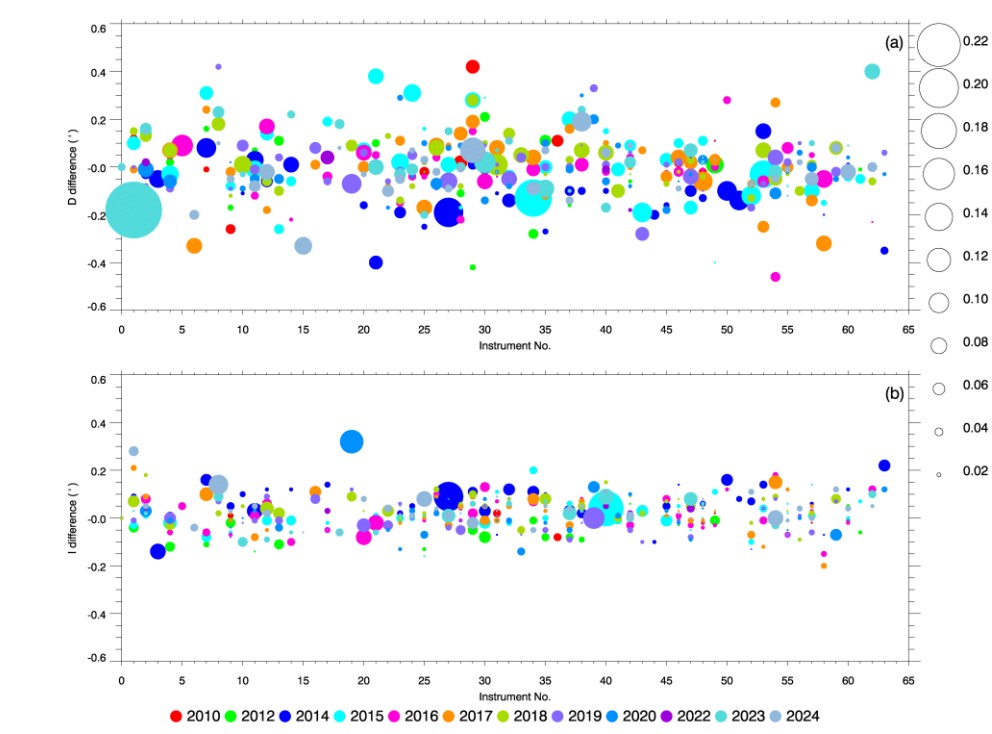

**Figure 3: Instrumental differences in declination D (a) and inclination I (b). The dots represent the instrument differences; the size of the dots is the standard deviation with the scale on the right. The colors indicate different years.**


First, the figures provide insights into instrument performance and operator proficiency. Small dots with large central values indicate significant instrumental differences, suggesting potential instrument malfunctions or operational issues by personnel. Particularly when the difference of D is relatively large while that of I is small, the most likely cause is the positioning error of the theodolite on the pillar, which also reflects technical shortcomings in observational practices. Conversely, large dots

with small central values signify dispersed data, which could arise from instrument related issues (e.g., unclear optical paths affecting reading accuracy) or inconsistent operational practices. More detailed explanations for tracking abnormal information can be found in He et al. (2009b). This graphical approach thus effectively monitors instrument performance and evaluates observational quality across operators.

Second, the collective distribution of all dots permits assessment of network wide data quality over multiple years. To quantify

network performance, we conducted statistical analyses of all instrumental differences. Statistical analysis of all differences (Fig. 4) reveals that declination (D) and inclination (I) measurements approximate normal distributions, with means of 0.00′ and 0.02′, and standard deviations of 0.13′ and 0.07′, respectively. Approximately 75.1% of declination and 86.8% of inclination measurements fall within ±1σ of the mean. When adopting a cumulative probability of 90% as the evaluation criteria for the entire network, the corresponding instrument differences thresholds are 0.21′ for D and 0.11′ for I, indicating

excellent consistency among network fluxgate instruments. Notably, declination measurements exhibit greater dispersion than

inclination values. This difference stems from the additional azimuth marker alignment required in declination measurements—a process more susceptible to operator error compared to inclination measurements. Another significant source of possible error in declination readings, which is not present in inclination readings, is the accuracy of setting (at 90° or 270°) on the vertical circle.

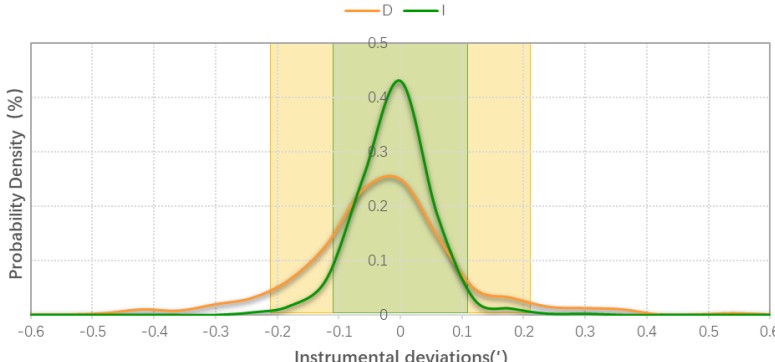


**Figure 4: The instrumental differences of declination D (orange line) and inclination I (green line).**

Furthermore, the dispersion of multiple dots corresponding to the same instrument also reflects its data quality and operation stability. Frequent personnel changes for same instrument have introduced operator dependent errors, manifesting as increased dispersion. To further explore the relationship between frequent personnel changes and the dispersion of instrument differences,

the frequency personnel change was defined as the ratio of non-repeated operators to the total number of comparative measurements for each instrument (from Table 1), serving as the x-axis, the dispersion degree was represented by the standard deviation of all instrumental differences for each instrument, serving as the y-axis. To enhance statistical significance, only instruments that participated in 3 or more comparisons were included in the analysis. As shown in Fig. 5(a), the dispersion degree of the instrumental differences increases with the frequent personnel changes, while this phenomenon is less

pronounced in Fig. 5(b). This indicates that frequent personnel changes increase observational errors and that personnel changes have a greater impact on D than on I. This result is consistent with the conclusion in the previous paragraph that D errors are larger than I errors. It further provides strong evidence supporting the explanation that operator errors primarily arise from the alignment of markers and the level adjustment of the theodolite.

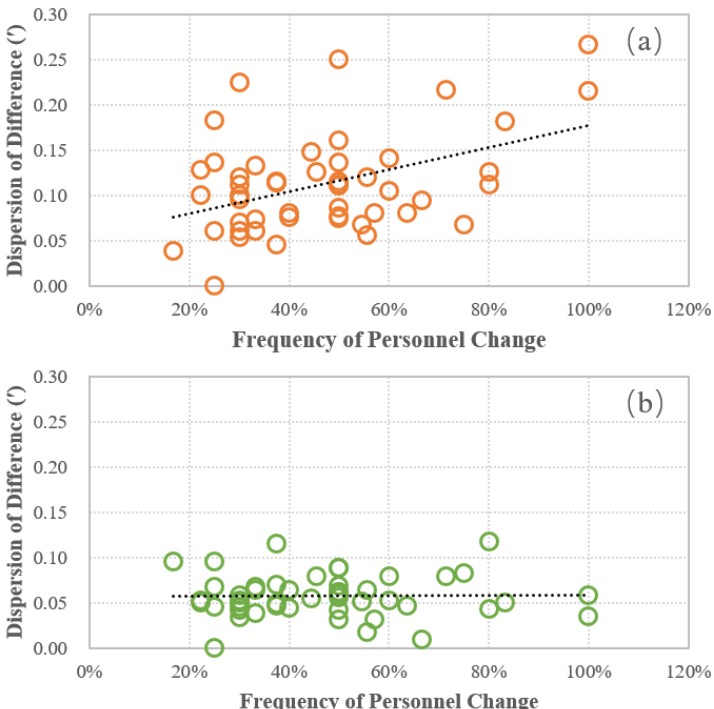

**Figure 5: The relationship between frequent personnel changes and the dispersion of instrument differences, (a) declination D, (b) inclination I, and the black dashed line is a linear fitting line.**

Additionally, the instrument differences of all 12 years' comparisons were classified to five group based on the instrument type, and the standard deviations were calculated for each group. This was done to simply compare the stability of observational results across different instrument types, as shown in Fig. 6. It can be clearly seen that MINGEO has better stability, followed

by Mag-01H and TDJ2E-NM, while the other two have relatively large dispersion, which is directly related to the resolution of the theodolite.

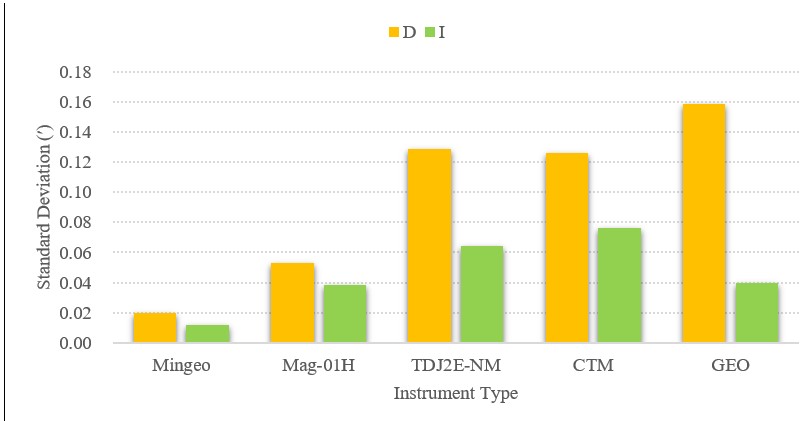

**Figure 6: The standard deviation of the instrument differences for five types of instruments.**

This simplified comparative measurements provide an efficient mechanism for identifying inter station differences, monitoring instrument performance, and ensuring standardized high quality observations across the network. However, the efficacy of this mechanism critically depends on the precision and accuracy of reference instruments. Beyond routine maintenance and calibration of reference standards, it is necessary to analyses long term stability and reliability. The following section will evaluate the reference instruments using all comparative measurement data through uncertainty analysis.

## 3 Robustness evaluation of the reference instrument

The robustness evaluation of the reference instrument requires quantification of its systematic deviation relative to the true values and associated uncertainties. The true value is the absolute, unbiased value of the measured physical quantity, which is typically unattainable directly in most cases. It can usually be represented by the arithmetic mean of sufficiently repeated measurement data, while uncertainty characterizes the dispersion of the measured value, indicating the range within which the

true value may lie, including Type A and Type B standard uncertainties. Type A uncertainty is a type of uncertainty evaluated through statistical methods (e.g., standard deviation of repeated measurement data) to assess the reliability and dispersion of measurement results. Its evaluation relies on the statistical analysis of repeated experimental data. While Type B uncertainty is based on non-statistical methods (e.g., instrument calibration certificates, empirical formulas, or known error limits), often combined with prior information or professional judgment (ISO/IEC GUIDE 98-3:2008). The assessment involves three key

components: determination of true values, characterization of the reference instrument's measurements, and statistical analysis of differences. This study evaluates 12 comparisons data covering 15 years from 2010 to 2024. First, the true values and uncertainties are assessed for each intercomparison session, followed by analyzing differences between the reference instrument and true values. Second, a temporal difference model is established using multi-year intercomparison results to evaluate the long term stability of the reference instrument. Finally, annual composite uncertainties are calculated by

integrating multi-year observational means and standard deviations from participating instruments, reflecting the dynamic characteristics of the measurement system. A comprehensive evaluation of the reference instrument's robustness is achieved through time series analysis and multi-source uncertainty synthesis.

### 3.1 Uncertainty analysis method for a single intercomparison

During the intercomparison process, fluxgate theodolites are employed to observe magnetic declination and inclination, with

baseline value comparisons serving as the final validation step. The analysis of uncertainties in this process encompasses several key error sources, including internal errors of the theodolite, repeatability errors, operator induced errors, and pillar correction errors. Environmental interference is excluded from consideration due to controlled laboratory conditions.

### 3.1.1 Uncertainty of fluxgate theodolite error

The fluxgate theodolite consists of two primary components: the theodolite and the fluxgate sensor. For the theodolite
component, the uncertainty of internal error of theodolite can be evaluated using the Type B standard uncertainty according to
ISO/IEC GUIDE 98-3:2008. Typically, the maximum permissible standard deviation of horizontal and vertical angles for one
measurement cycle is considered as the standard for theodolite level classification based on GB/T 3161-2015 (Standardization
Administration of China, 2015). This can be used as a parameter to evaluate the theodolite. Therefore, the first step is to consult
the theodolite's relevant manual and obtain its parameter. Assuming these maximum permissible standard deviations follow a
normal distribution, the Type B standard uncertainty for theodolites can be calculated using Eq. (8) from ISO/IEC GUIDE 98-
3:2008.

$$u_{b1} = \delta/3, \tag{8}$$

where $\delta$ is the maximum permissible standard deviation. Table 2 provides the relevant information and parameters of these
five fluxgate theodolites.

**Table 2: The relevant information and parameters of the five fluxgate theodolites**

| Instrumnent | Theodolite | | | Sensor | | |
|---|---|---|---|---|---|---|
| | Model | Resulution | Maximum permissible standard deviation | Model | Resulution | Offset |
| MinGeo-DIM | Theo 010A | 1″, estimation 0.1″ | ≤ ±2″ | Model G | 0.1 nT | ±1nT |
| Mag-01H | T1 | 6″, estimation 3″ | ≤ ±3″ | Mag A | 0.1 nT | ±1nT |
| TDJ2E-NM | TDJ2E | 1′, estimation 6″ | ≤ ±6″ | Mag A | 0.1 nT | ±1~5nT |
| CTM-DI | CJ6 | 1′, estimation 6″ | ≤ ±4″ | — | 0.1 nT | ±1nT |
| GEO-DI | J6 | 1′, estimation 6″ | ≤ ±6″ | — | 0.1 nT | ±1nT |

For the fluxgate sensor, errors arise from the liquid crystal display's limited resolution ($\varepsilon$=0.1 nT), which follows a uniform
distribution. The corresponding Type B standard uncertainty is derived via Eq. (9) given in ISO/IEC GUIDE 98-3:2008:

$$u_{b2} = \varepsilon/\sqrt{3}, \tag{9}$$

where $\varepsilon$ is the limited resolution.

In addition, there may be other uncertainties that affect the observations, such as magnetic contamination of the theodolite
body. Although these effects are difficult to quantify, they increase the uncertainty of the measurement results and will also be
reflected in the measurement results. So the synthesized internal uncertainty for each instrument is computed as the root sum
square of two elements, as shown in Eq. (10):

$$u_{inst,i} = \sqrt{u_{b1,i}^2 + u_{b2,i}^2} \tag{10}$$

### 3.1.2 Uncertainty of repeatability and operator error

During an instrument comparison, operator from different observatories use their own instruments to perform 6~8 repeated observations (i.e., 6~8 sets of results). Based on these results, the standard deviation $(s_{rep,i})$ and associated Type A uncertainty $(u_{rep,i})$ of the repeatability error for each operator-instrument-combination can be calculated using Eq.(11):

$$u_{rep,i} = \frac{s_{rep,i}}{\sqrt{N}}, \tag{11}$$

$$s_{rep,i} = \sqrt{\frac{1}{N-1}\sum_{k=1}^{N}\left(x_{i,k} - \frac{1}{N}\sum_{k=1}^{N}x_{i,k}\right)^2}, \tag{12}$$

where $x$ is the baseline value calculated according to Eq. (4) and corrected for pillar difference, $N$ is the number of baseline values for each instrument. instruments involved in the comparison.

Since multiple operator-instrument-combinations from different observatories are involved in an instrument comparison, the
standard deviation $(s_{between})$ of all operator-instrument-combinations results can be calculated following Eq. (13). It includes both the internal error of theodolite and the operator error, so the operator uncertainty $(u_{oper})$ can be derived by subtracting the averaged instrumental uncertainties, as shown in Eq. (14):

$$s_{between} = \sqrt{\frac{1}{N-1}\sum_{i=1}^{N}(\overline{x}_i - \overline{\overline{x}})^2} , \qquad \overline{\overline{x}} = \frac{1}{N}\sum_{i=1}^{N}\overline{x}_i, \tag{13}$$

$$u_{oper} = \sqrt{s_{between}^2 - \frac{1}{N}\sum_{i=1}^{N}u_{inst,i}^2} , \tag{14}$$

where $\overline{x}$ is the average baseline value of each instrument, $N$ is the number of instruments involved in the comparison.

### 3.1.3 Uncertainty of pillar correction error

Pillar correction errors are incorporated based on pillar difference measurements. In comparison, different instruments may be installed on different observation pillars, and magnetic field gradients may exist between pillars (i.e., pillar differences). To unify the observation results of these instruments to the standard pillar for comparison with the standard instrument's results,
pillar difference corrections must be applied to the measurement data of each instrument. So the influence of pillar difference errors must be considered in the uncertainty analysis. These pillar differences $(\Delta W_{SO})$ and their uncertainty $(u_{pier})$ can be obtained from the measurement results of pillar differences at each observatories. They are obtained by using repeated measurement data and calculating according to Eq. (6) or Eq. (7). They can also be checked using all the comparison data.

Given that the magnetic gradient within the observation room is very small, the pillar difference is therefore typically minimal.
Nevertheless, prior to initiating each comparison process, it was remeasured to ensure accuracy. Table 3 presents the pillar differences and their uncertainties at the observatory where the comparison is conduced.

Table 3: Pillar differences and their uncertainties

| Pillar No. | Pillar Difference $(\Delta W_{SO})$ | | Uncertainty $(u_{pier})$ | |
|---|---|---|---|---|
| | D(′) | I(′) | D(′) | I(′) |

| | | | | |
|---|---|---|---|---|
| **1#** | 0.00 | 0.00 | 0.01 | 0.01 |
| **2#** | 0.06 | -0.03 | 0.05 | 0.02 |
| **3#** | 0.10 | 0.15 | 0.04 | 0.03 |
| **4#** | 0.17 | 0.07 | 0.03 | 0.02 |
| **5#** | 0.13 | 0.28 | 0.07 | 0.04 |
| **6#** | 0.30 | 0.11 | 0.03 | 0.02 |

### 3.1.4 The total synthesized uncertainty

The total synthesized uncertainty for each instrument is then aggregated as the root sum square of all contributing factors, according to Eq. (15):

$$u_i = \sqrt{u_{inst,i}^2 + u_{oper}^2 + u_{rep,i}^2 + u_{pier,i}^2} \tag{15}$$

Finally, the ensemble mean ($\mu_{group}$) and combined uncertainty ($u_{group}$) for all instruments are computed using a weighted average approach. Weights ($\omega_i$) are assigned inversely proportional to the square of each instrument's total uncertainty,

ensuring higher precision instruments exert greater influence, using Eq. (16) and (17):

$$\mu_{group} = \frac{\sum_{i=1}^{N} \omega_i \mu_i}{\sum_i^N \omega_i} \quad , \quad \omega_i = \frac{1}{u_i^2} \tag{16}$$

$$u_{group} = \frac{1}{\sqrt{\sum_i^N \omega_i}} \tag{17}$$

This comprehensive methodology transforms the intercomparison into a robust experiment integrating multi operators' collaboration, parallel instrumentation, and repeated measurements, ensuring rigorous uncertainty quantification.

**3.2 Multi years intercomparison analysis method**

Building on the uncertainty analysis of single intercomparison sessions, this section evaluates the long term stability and robustness of the reference instrument using data accumulated over 12 comparisons within the GNC. Each intercomparison session involves comparing the reference instrument, mounted on a standardized pillar, against the ensemble results of participating instruments. Since the reference instrument requires no pillar correction, its mean value ($\mu_s$) and associated

uncertainty ($u_{rep,s}$) are derived from 6~8 repeated measurements per session. The repeatability standard deviation ($s_{rep,s}$) and corresponding uncertainty ($u_{rep,s}$) are calculated using Eq. (18) and (19):

$$u_{rep,s} = \frac{s_{rep,s}}{\sqrt{N}} \tag{18}$$

$$s_{rep,s} = \sqrt{\frac{1}{N-1}\sum_{k=1}^{N}\left(x_{s,k} - \mu_s\right)^2} \quad , \quad \mu_s = \frac{1}{N}\sum_{k=1}^{N} x_{s,k} \tag{19}$$

The total uncertainty of the reference instrument ($u_s$) incorporates its internal error ($u_{inst,s}$), the operator induced uncertainty ($u_{oper}$), and repeatability uncertainty, as expressed in Eq. (20):

$$u_s = \sqrt{u_{inst,s}^2 + u_{oper}^2 + u_{rep,s}^2} \qquad (20)$$

The difference ($\Delta$) between the reference instrument and the weighted ensemble mean ($\mu_{group}$) of all participating instruments, along with its uncertainty ($u_\Delta$), is quantified for each session using Eq. (21) and (22):

$$\Delta = \mu_{group} - \mu_s \qquad (21)$$

$$u_\Delta = \sqrt{u_s^2 + u_{group}^2} \qquad (22)$$

A consistency criterion ($|\Delta| \leq 2u_\Delta$) is applied to verify agreement at a 95% confidence level.

To assess long term stability, differences ($\Delta$) from $M$ years intercomparison sessions are compiled into a time series plot, enabling visual detection of potential drifts caused by environmental fluctuations or instrumental aging. The multi-year mean difference ($\overline{\Delta}$) and its uncertainty ($u_{\overline{\Delta}}$) are calculated through Eq. (23) and (24):

$$\overline{\Delta} = \frac{1}{M}\sum_{m=1}^{M}\Delta_m \qquad (23)$$

$$u_{\overline{\Delta}} = \sqrt{\frac{1}{M}\sum_{m=1}^{M}u_{\Delta,m}^2 + \left(\frac{s_\Delta}{\sqrt{M}}\right)^2} \qquad (24)$$

Here, ($s_\Delta$), represents the standard deviation of $\Delta$ across all sessions. The final robustness criterion ($|\overline{\Delta}| \leq 2u_{\overline{\Delta}}$) ensures the reference instrument's performance remains within acceptable bounds over extended periods. This integrated approach combines temporal trend analysis with uncertainty propagation, providing a comprehensive evaluation framework for maintaining measurement integrity in long term geomagnetic monitoring.

### 3.3 Results analysis

Using the methodology described above, we analyzed 12 years of instrumental difference data for declination (D) and inclination (I). The time series of mean differences ($\Delta$) between the reference instrument and all tested instruments, along with their uncertainties ($u_\Delta$), are shown in Fig. 7. In the figure, orange and green histograms represent mean differences for D and I, respectively, while curves of corresponding colors indicate twice the uncertainty ($2u_\Delta$). According to the criterion $|\overline{\Delta}| \leq 2u_{\overline{\Delta}}$, most mean differences fall within the $2u_\Delta$ range.

However, mean differences for both D and I exceeded this threshold in 2014. A retrospective review of the raw data revealed no definitive cause for this anomaly. Notably, 2014 involved an observational training program where measurements were conducted by inexperienced personnel, and the definite uncertainty was notably low. This suggests potential transient impacts on the reference instrument. Additionally, the mean difference for D in 2018 slightly exceeded $2u_\Delta$, though no conclusive explanation has been identified. Nevertheless, the long term mean difference data demonstrate that the reference instrument has maintained stable operation and reliable performance throughout the study period.

Finally, using Eq. (23) and (24), we evaluated the reference instrument's long term stability by calculating the multi years average mean differences and their uncertainties. For declination (D), the average mean difference was $\overline{\Delta_D} = -0.004'$ with an uncertainty of $u_{\overline{\Delta D}} = 0.054'$. For inclination (I), the values were $\overline{\Delta_I} = 0.022'$ and $u_{\overline{\Delta I}} = 0.023'$. Applying the criterion $|\overline{\Delta}| \leq 2u_{\overline{\Delta}}$ (95% confidence level), both D and I meet this requirement, confirming the reliability of the reference instrument's long term observational data.

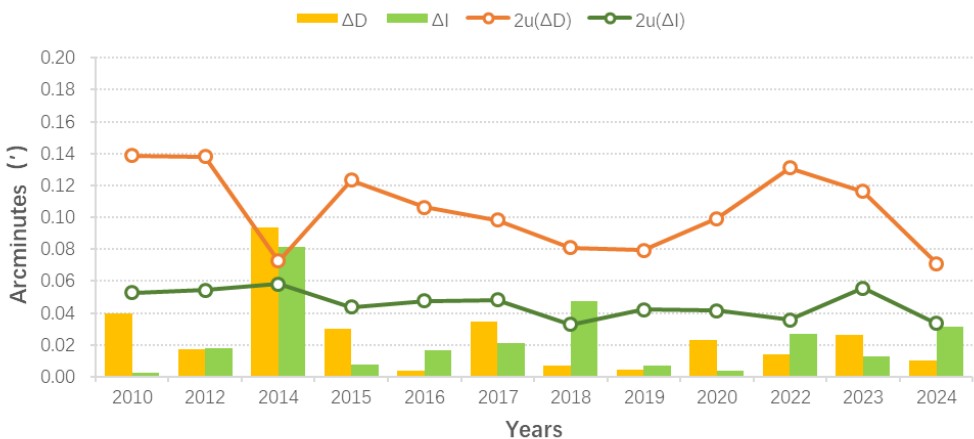

**Figure 7: Time Series of Mean Differences and Uncertainties Between Reference and Tested Instruments.**

Operator dependent errors (human differences) during comparative measurements were calculated using Eq. (14), as illustrated in Fig. 8. The light orange and light green filled areas represent the differences for D and I, respectively. Results show consistently higher operator dependent errors in declination measurements compared to inclination, with D errors persistently exceeding I values. This difference arises from the additional azimuth marker alignment step, and the accuracy of the vertical circle setting (at 90 °or 270 °) required for declination measurements, which introduces greater operator variability. The mean operator dependent errors were 0.13′ for D and 0.06′ for I, aligning closely with experimental results(0.18′ for D and 0.08′ for I)reported by He (2019a), thereby validating the methodology's effectiveness in quantifying human induced errors.

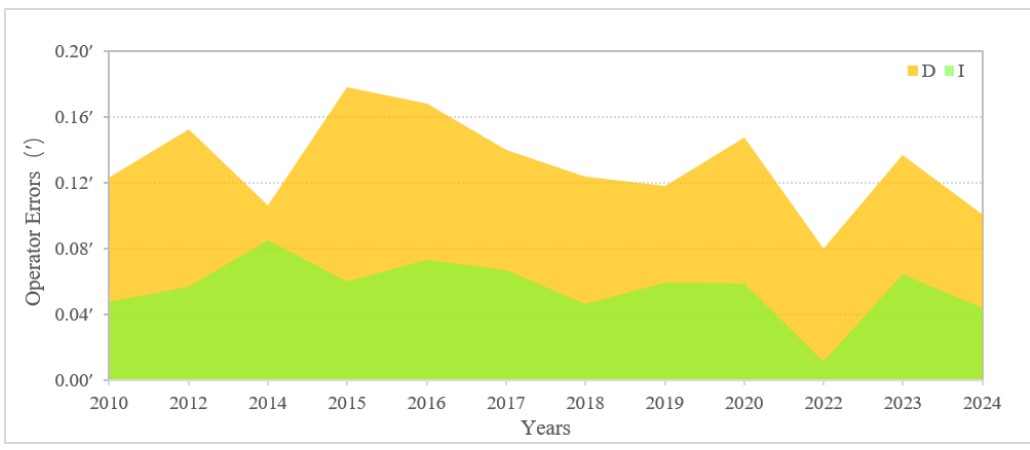

Figure 8: Operator dependent Errors in Declination (D) and Inclination (I) Measurements.

## 4 Conclusion and outlook

This study systematically analyzed 12 years of comparative measurement data from the GNC. The results underscore the critical importance of instrument comparisons in geomagnetic observatory networks, demonstrating their dual role in (i) monitoring instrument performance and evaluating data quality at network scales, and (ii) providing researchers with transparent insights into the accuracy of absolute measurements. These assessments also offer valuable references for evaluating the reliability of scientific conclusions derived from geomagnetic data.

A comprehensive evaluation framework was developed to assess the robustness of reference instruments. By decomposing uncertainties into instrument specific errors, repeatability errors, operator dependent errors, and pillar correction errors during individual comparisons, we achieved precise quantification of multi sources uncertainties. The weighted mean method enhances result reliability by prioritizing high precision instruments. The integration of multi instruments datasets through weighted averaging significantly improved result reliability. By constructing time series of mean differences (Δ) and their uncertainties between the reference instrument and tested instruments, we analyzed the long term stability of the reference instrument.

Furthermore, multi-year data analysis revealed the mean differences and inter-annual composite uncertainties, while significance tests based on mean differences and expanded uncertainties provided objective criteria for instrument performance, ensuring measurement consistency in complex environments. Additionally, our methodology successfully separated operator dependent errors from inherent instrument errors, with computational results aligning closely with experimental validations. However, this approach has limitations. For instance, it overlooks operator instrument interactions and environmental factors (e.g., humidity fluctuations), which may introduce systemic biases. Long term stability analysis also requires extensive multi years comparative data to ensure statistical power, limiting rapid field applications. Future research should focus on model optimization, such as incorporating environmental sensor data to establish temperature/humidity compensation mechanisms

or developing automated tools to streamline multi sources uncertainty synthesis. With continuous refinement, this methodology holds promise for advancing standardization and long term stability in geomagnetic observation networks.

Furthermore, geomagnetic observatories serve as primary facilities for measuring the secular variation of the Earth's magnetic field. The measurement accuracy for directional elements (e.g., declination and inclination) is typically required to be below 0.1', while the accuracy for intensity elements (e.g., total field strength) should be within 1nT. Modern instruments, such as the Zeiss 010B fluxgate theodolite, theoretically possess sufficient precision to achieve these targets. However, in practice, attaining such accuracy remains challenging due to various sources of error, particularly operator dependent errors. With advancements in automation and the global adoption of high precision instruments (Rasson et al.,2011; Gonsette et al., 2017; Hegymegi et al.,2017) in the future, it is anticipated that operator dependent differences will be eliminated, thereby obtaining higher quality geomagnetic observational data.

### Data availability

The raw data are available upon request from the corresponding author at darcyli@163.com.

### Author contribution

YH and QL initiated the study and designed the analysis methods. XZ and FY carried them out. SZ analyzed the data and results. YH prepared the manuscript with contributions from all coauthors.

### Competing Interests

The authors have no competing interests to declare.

### Acknowledgements

The authors extend sincere gratitude to all colleagues involved in the instrument comparisons, whose dedicated efforts were pivotal to the realization of this research.

### Funding Information

Supported by National Key R&D Program of China (2023YFC3007404); National Natural Science Foundation of China (42374092); DI Magnetometer Comparison (0525205).

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
