# Peer review of "Evaluating data quality and reference instrument robustness: insights of 12 years DI magnetometer comparisons in the Geomagnetic Network of China"

_EGUsphere, 2025_

## Author Comment (AC1)

**Responds to the comments 1**

The author sincerely appreciates the reviewers' meticulous evaluation. In response to the constructive suggestions provided by the reviewers, we have made revisions in the revised manuscript. Regarding errors in textual expression, we have carefully corrected them. All modified content has been highlighted in red font in the revised manuscript. Below are the specific responses to the reviewers' comments.

Comments

The subject of geomagnetic absolute instrument comparisons across national and international observatory networks is an important issue worthy of the exploration presented in this paper. The paper presents useful introductory information and explanation on the theory and methods used for the analysis together with specific details on the observatories and instrumentation from the Chinese network used in the study. Methods of calculation and statistical analysis of results from the multi-year data set are presented effectively in diagrams and discussion.

Line 92 ; The meaning of the word "integration" in this context is not clear to me.

**Responds:** The term "integration" here is intended to convey the meaning of "joint use". It has been modified to "By using the geomagnetic declination (D), inclination (I), and the total magnetic intensity (F) measured by the proton magnetometer, all the absolute components of the Earth's magnetic field can be calculated. This will facilitate the subsequent baseline calculations of variometer for all components (such as east, north, and vertical directions)."

A map of the observatory locations would be interesting, but obviously not necessary for the arguments presented in the paper.

**Response:** Following this suggestion, a map of the observatory locations has been added in the revised draft.

I suggest a more precise description of the instruments would be useful in Table 2, including both the theodolite make/model and also the fluxgate make/model.

**Response:** Thank you for your suggestion. We have added an introduction regarding theodolites and fluxgate sensors in the revised manuscript. More detailed information about these instruments has been added in Table 2, including models, resolution, maximum, etc. Information about the observers has also been added to Table 1.

Line 175:   A reference would be helpful for Type B (line 175 ) and Type A (line 190) uncertainties

**Response:** The true value can usually be represented by the arithmetic mean of sufficiently repeated measurements, while the uncertainty includes Type A and Type B standard uncertainty (ISO/IEC ,2008).

International Organization for Standardization and International Electrotechnical Commission: Uncertainty of measurement, Part 3: Guide to the expression of uncertainty in measurement (GUM:1995), ISO/IEC GUIDE 98-3:2008, https://www.iso.org/resources/publicly-available-resources.html (last access: 2 September 2025), 2008.

Line 180: Table 3: Is it possible to relate the "Class" column in Table 3 to instrument types listed in Table 2?

**Response:** This is an excellent suggestion. We have added a column for instrument type in Table 3 (it is table 2 in the revised draft).

Corrections:

Include the word "of" in the title "... from 12 years of DI magnetometer..."

Line 69: replace "..A of the marker given," with "... of the marker is known,..."

Line 70: "(with the vertical circle maintained at ...")

Line 75: "Inclination measurements follows analogous procedures in the vertical plane, omitting azimuth mark referencing and using the magnetic meridian derived from the preceding declination measurements"

Line 80: "The declination measurement protocol is preceded and followed by sensor up and down azimuth mark readings and then involves four configurations...."

Line 83 : "D'=(Dz + D2 + D3 + D4)/4 and substituting the value D' into equation (1)

Line 96: "...systematic verification of inter-instrument differences across..."

Line 107: "Where, W0(i:j) ,,,,"

Line 110: "... of inter-pillar differences..."

Line 122: Table 1:

Line 125: Table 2: Correct the spelling of "Instrument" in table header

Line 152: "susceptible to operator error compared to ...."

Another significant source of possible error in declination readings, which is not present in inclination readings, is the accuracy of setting (90/270) on the vertical circle.

Line 158: remove "its" ..."necessary to analyses long term stability ...."

Line 163: "... 12 years of intercomparison ..."

Line 166: "...using multi-year..."

Line 167: "...integrating multi-year..."

Line 169: "... and multi-source uncertainty..."

Line 193: "analyzing inter-station variances"

Line 226: "The multi-year mean..."

Line 240: "However, mean differences for both D and ..."

Line 258: "...azimuth mark alignment step, and the accuracy the vertical circle setting (90/270), required for declination measurements

Line 277: "Furthermore, multi-year data analysis revealed the mean differences and inter-annual...."

Line 293: the use of the phrase "... and so on" seems inappropriate

The words "discrepancy" (lines 96, 114, 100) "difference" (line 116), "deviation" (figure 2), are used in different places throughout the text. I suggest consistent use of "difference" would be better.

**Response:** The author fully accepts the upper suggestions (the title, L69, L70, L75, L80, L83, L96, L107, L110, L122, L125, L152, L158, L163, L166, L167, L169, L193, L226, L240, L258, L277, L293 in the original manuscript) and has made corresponding modifications at the relevant positions. The word "discrepancy" and "deviation" in the entire text have also been replaced with "difference".

---

## Author Comment (AC2)

**Responds to the comments 2**

Thank you very much for the reviewer's careful review. Your suggestions have greatly helped enrich the content and improve the quality of our article. Below are detailed modifications and response to your suggestions. In the revised manuscript, we also marked it in red font.

General Comments

The paper analyses and discusses the results of DIM inter-comparison campaigns conducted between 2010 and 2024, involving 46 GNC-operated geomagnetic observatories in China. Following a concise introduction to the instrumentation and measurement principles, the authors enumerate the sources of error.  The introduction has some shortcomings and needs to be complemented as detailed below.

Although the focus of the present work is on error analysis, the authors did not utilise the error information that can be derived from the absolute measurement sequences recorded in the measurement protocol sheets, such as sensor misalignment errors or electronic offset-related errors (see, e.g., Csontos and Sugar, 2024). These parameters can only be inferred from the measurement sequences (e.g., D1, D2, D3, D4) but not from the derived baseline values that underpin this study. This may explain their omission in the discussion, given that the protocols were not available. It is strongly recommended that the authors extend the error analysis to define and evaluate these parameters in future work. Additionally, there are some unclear points in the paper and missing information that require clarification.

Csontos, D. Sugar (2024), Dataset of geomagnetic absolute measurements performed by Declination and Inclination Magnetometer (DIM) and nuclear magnetometer during the joint Croatian-Hungarian repeat station campaign in Adriatic region, Data in Brief 54,110276, doi.org/10.1016/j.dib.2024.110276.

**Responds:** The theodolites are high-precision instruments, but they inevitably contain certain errors, such as misalignment errors between the mechanical axis of theodolite, the optical axis of the telescope, and the magnetic axis of the fluxgate sensor; collimation errors; non-orthogonality errors of the horizontal and vertical axes; uneven graduation errors of the reading circle; index errors; and errors caused by non-zero electronic offsets, which prevent accurate determination of magnetic declination and inclination from a single reading of the horizontal/vertical circle (Lauridsen, 1985; Newitt et al., 1996; Csontos and Sugar, 2024). However, in theory, most of these errors can be eliminated through the four position measurement process, and some of them (two misalignment errors between the fluxgate sensor axis and the optical axis of the telescope in the horizontal/vertical planes, and the offset error of the fluxgate sensor) can be calculated from the measurement results (Bitterly et al., 1984). Nevertheless, errors cannot be completely eliminated and will still exist, which is the main reason for the differences between different instruments and the source of uncertainty in measurement results. The instrument differences defined in this paper are the comprehensive differences of the entire instrument system, representing the differences between results obtained by the instruments after four measurement processes, under the assumption of no personnel operation error. Consequently, this article does not separately explore the impact of

various internal errors on measurement results, but rather takes their combined effects as the overall intrinsic errors of the theodolite. In future work, we will try to expand error analysis and evaluate the impact of system errors or parameters within the instrument system based on your suggestions. Thank you for your suggestion.

Specific comments

l 9 12 years: between 2010 and 2024
**Responds:** This sentence has been modified to "A statistical analysis was conducted on 12 years of geomagnetic instrument comparison data from the Chinese Geomagnetic Network (GNC) between 2020 and 2024"

l10 applying a 90% threshold: threshold for what?
**Response:** The threshold in the original text refers to the cumulative probability, which is intended to indicate when the probability density of instrument differences accumulates to 90%. Now, in order to express more clearly, the "threshold" in the original abstract and line 148 has been modified and replaced with "cumulative probability". The original sentence has been modified to "The study reveals that when the probability density of instrument differences accumulates to 90%, the corresponding instrument deviation are 0.21 ′ (D component) and 0.11′ (I component)"

l18 due to the complexity of azimuth alignment > due to the complexity of azimuth alignment and levelling
**Response:** The sentence has been modified.

l27 discrepancies > differences
**Response:** The word has been corrected.

l40 missing spaces following commas
**Response:** The error has been corrected.

l59 fluxgate sensor, mounted coaxially with > fluxgate sensor, mounted parallel to (the sensor cannot be mounted coaxially with the optical axis). This positioning is also a potential source of error not mentioned in the paper. It implies that the observations are made
**Responds:** Following this suggestion, an introduction about the error source of the theodolite has been added to the revised section 2.3. The specific content is as follows:
The theodolites are high-precision instruments, but they inevitably contain certain errors, such as misalignment errors between the mechanical axis of theodolite, the optical axis of the telescope, and the magnetic axis of the fluxgate sensor; collimation errors; non-orthogonality errors of the horizontal and vertical axes; uneven graduation errors of the reading circle; index errors; and errors caused by non-zero electronic offsets, which prevent accurate determination of magnetic declination and inclination from a single reading of the horizontal/vertical circle (Lauridsen, 1985; Newitt et al., 1996; Csontos and Sugar, 2024). However, in theory, most of these errors can be eliminated through the four position measurement process, and some of them (two misalignment errors between the fluxgate sensor axis and the optical axis of the telescope in the horizontal/vertical planes, and the

offset error of the fluxgate sensor) can be calculated from the measurement results (Bitterly et al., 1984). Nevertheless, errors cannot be completely eliminated and will still exist, which is the main reason for the differences between different instruments and the source of uncertainty in measurement results. The instrument differences defined in this paper are the comprehensive differences of the entire instrument system, representing the differences between results obtained by the instruments after four measurement processes, under the assumption of no personnel operation error. Consequently, the text does not explore the impacts of various internal errors on the measurement results. Consequently, the impact of various internal errors on measurement results is not separately explored in this article, but their combined effects are considered as the overall internal error of the theodolite.

This positioning is also a potential source of error. For clearer expression, some modifications have been made to the original text. The difference caused by positioning errors were observed during the comparison, specifically cases where the difference of D is relatively large while that of I is small.

l60 it generates zero output: assuming zero offset

**Response:** This sentence has been corrected.

l65 vertical > (magnetic) meridional: Inclination measurements are carried aligning the instrument with the magnetic meridian determined through declination measurements.

**Response:** This sentence has been corrected.

l67 Two observations are needed to find the true north direction. One with sensor up and another with sensor down to eliminate errors associated with the optical misalignment of the theodolite.

**Responds:** In the revised manuscript, we have added two observations regarding the sensor up and down. These sentences are "In order to eliminate errors associated with the optical misalignment of the theodolite, two observations are required to find the true north direction, one with sensor up and the other with sensor down. Finally, the direction of the azimuth marker can be determined through two readings and recorded as M.

l75 vertical > (magnetic) meridional

**Response:** The word has been corrected.

l75 omitting azimuth marker: the vertical reference is provided by the gravity field through the suspension system of the theodolite.

**Responds:** The description of vertical reference has been added to the revised manuscript. The supplementary sentence is "Inclination measurements follows analogous procedures and is carried out in the magnetic meridional plane derived from the previous declination measurements, while also within the vertical reference provided by the gravity field through the theodolite suspension system. "

l79 followed > follows

**Response:** The word has been corrected.

l80 four configurations > two azimuth readings and declination observations in four different positions to eliminate errors associated with theodolite optics, sensor misalignment and electronics offset.

**Responds:** The original text has been revised based on this suggestion, and the revised sentence is "The declination measurement protocol is preceded and followed by sensor up and down azimuth marker readings and then involves four configurations: (i) telescope East/sensor up ($D_1$), (ii) telescope West/sensor down ($D_2$), (iii) telescope East/sensor down ($D_3$), and (iv) telescope West/sensor up ($D_4$). Four different position observations can eliminate errors associated with theodolite optics, sensor misalignment and electronics offset. Then final declination value is derived through arithmetic averaging:"

l83 Eq. (2) misses a +/-90° term.

**Response:** Eq.(2) has been corrected.

l88 Two distinct: start a new paragraph here

**Response:** New paragraph has been started.

l92 Integration of declination: start a new paragraph here

**Response:** New paragraph has been started.

l96 discrepancies > differences [not only here but several times later]

**Response:** The term 'difference' in this article has been replaced with 'difference'.

l100 under stable operation > be more specific about the stable operational conditions (temperature, magnetic cleanliness, etc.)

**Responds:** Thanks for this suggestion. A description of stable operating conditions has been added in the text. The specific content is "Modern variometers exhibit high precision performance with quasi constant baseline characteristics under stable operating conditions, while underground observation rooms of geomagnetic observatories (far from cities or villages) can provide such operating conditions, including no influence of magnetic objects, low electromagnetic background noise, indoor annual temperature variation not exceeding 10 ℃, daily variation not exceeding 0.3 ℃, and so on. "

l107 Were, > , where
**Response:** The word has been corrected.

l107-108 Clarify the relation between minutes i-j and k.
**Responds:** The relation between minutes i-j and k has been clarified in the text. The content are ",Where $(i:j)$ is the time interval (typically minutes) for measurement, $(k)$ is the k-th time, the average time of interval $(i:j)$, $W_o(i:j)$ is the absolute field value for the time interval $(i:j)$, $W_R(k)$ is the variometer recorded value at time $k$, and $W_B(k)$ is the derived baseline value."

l109 across distinct pillars > on different pillars
**Response:** The term has been corrected.

l106 and 111: It is a bit confusing that the argument of W_B is time in Eq. (4) but location in Eq. (5). Be consistent.

**Responds:** This is a very important proposal. The expressions of formula (4) and formula (5) are indeed inconsistent. The expression of formula (5) is inappropriate. For clarity, we have moved the symbols "s/o" representing different pillars in formula (5) to the subscript of W instead of writing them in parentheses.

$$\Delta U_{SO} = W_{BS} - W_{BO} + \Delta W_{SO}$$

l112 Where, > , where

**Response:** The word has been corrected.

l113 Some notes on how the inter-pillar difference is derived would be beneficial.

**Responds:** The description of pillar difference, measurement methods and calculation formulas have been added in section 2.4 of the revised manuscript. And the specific pillar differences and their uncertainties of the observatory for instrument comparison were provided in revised section 3.1.3.

l115 cross observatory fluxgate theodolite comparisons: Mentioning observatories in this context is a bit confusing to me. Would not it be better to say simply „cross-comparisons of fluxgate theodolites"?

**Responds:** This sentence has been corrected to "This methodology enables cross comparisons of fluxgate theodolite through pillar reference baseline correction."

Table 1: Some information on the location of the observatories would be beneficial.

**Responds:** Table 1 shows the locations and times of the comparison work. According to the suggestions of other reviewers, this table is not very relevant to the research content, so it has been deleted. But the distribution map of all observatories has been added in the text, as shown in Figure 2.

Table 2: Some more detailed information on the instruments (type and angular resolution of the theodolites, type of the magnetic sensor/electronics) would be beneficial if this is available to the authors. There is not any information on the observers. One can only assume that all instruments were operated by different individuals, and the same person across different years. However, this is not necessarily the case.

**Response:** Thank you for your suggestion. We have added an introduction regarding theodolites and fluxgate sensors in the revised manuscript. More detailed information about these instruments has been added in Table 2, including models, resolution, maximum, etc. Information about the observers has also been added to Table 1.

l127 deviations > differences

**Responds:** The deviations related to instrument and pillars in the entire text have been replaced by differences.

l130 scaled according to the legend on the right: There are dots in the figure obviously larger than the largest shown in the referred scale.

Responds: The right legend indicates the dot size corresponding to its value. If the dot in the figure is larger than one dot in the legend but smaller than another dot, it indicates that the dot's value in figure is between the values corresponding to the two dots in the legend. Following this suggestion, several numerical legends have been added to the right legend to better correspondence with the dots in the figure.

Figure 2 Units are missing both from the y-axis labels and the scale shown in the legend.

**Responds:** The units of numerical values have been added in Figure 2 (now Figure 3).

Figure 2 There are 46 categories along the x-axis. This is equal to the number of observatories but not the number of the instruments. The entire paper focuses on instrumental differences, yet this crucial figure combines and merges the various instruments. This needs to be corrected.

**Responds:** Thank you very much for this suggestion. Figure 2 (now Figure 3) has been redrawn in the revised draft, and the categories along the x-axis are no longer depends on observatories but on the number of instruments.

l137 centring errors: What do you mean on centring errors? Positioning accuracy of the theodolite on the pillar? This has effect primarily on the declination baseline differences but small if any on inclination baseline differences. Have you checked this to find the reason of the differences?

**Responds:** Yes, the centing errors here is the positioning accuracy of the theodolite on the pillar. For clearer expression, some modifications have been made to the original text. The difference caused by positioning errors were observed during the comparison, specifically cases where the difference of D is relatively large while that of I is small. We checked the instrument's condition, compared its previous comparison results, referenced with those of the synchronously comparison instruments to ensure no errors occurred in the standard instrument's observation. After repositioning and leveling adjustments, subsequent measurements showed minimal difference compared to the standard instrument's results. Therefore, such cases do indeed exist.

l136-140 There are also large dots with large central values. Some more detailed analysis of the obtained differences would be beneficial here. Some conclusion, e.g. on the accuracy of various instrument types. Typical sources of error, etc.

**Responds:** According this suggestion, some new analysis of the obtained difference were conducted, and two paragraphs and two new figures were added to describe the accuracy of various instrument types and the typical sources of error.

l140 dispersion of multiple dots corresponding to the same station: This is where various instruments belonging to the same stations are mixed up. This needs to be corrected or the interpretation, statements and conclusions need to be corrected.

**Responds:** Following this advice, figure 2 (now Figure 3) has been redrawn in the revised draft, and the categories along the x-axis are no longer depends on observatories but on the number of instruments. So the dispersion of multiple dots corresponding to the same instrument. The corresponding interpretation has been corrected.

l141 Frequent personnel changes: This obviously has a great effect on the results. Information on observers are totally missing. (They could be identified e.g., by two numbers: 1st for the observatory, 2nd for the individual). Without having this information some of the statements (e.g., „This graphical approach thus effectively monitors instrument performance") must be refined.

**Responds:** Thank you very much for this suggestion. Table 1 has been reorganized based on the participation instrument, while the number of times each instrument participated in comparison and the number of personnel operating it (with non-repeated counts) have been also list. To further explore the relationship between frequent personnel changes and the dispersion of instrument differences, new figures (Figs. (5) and (6)) has been added in the revised manuscript. The frequency personnel change was defined as the ratio of non-repeated operators to the total number of comparative measurements for each instrument, serving as the x-axis, the dispersion degree was represented by the standard deviation of all instrumental differences for each instrument, serving as the y-axis. So the frequent personnel changes are visible. Thus, to a certain extent, it can reflect the impact of frequent personnel changes.

l146-147 "means of 0.00′ and 0.02′" AND "indicating excellent consistency among network fluxgate instruments": These values depend on the choice of the reference instrument. Please clarify how the choice as made.

**Responds:** The means of "0.00′ and 0.02′" indicates that the reference instrument has excellent consistency with the fluxgate instruments among network. The selection of this reference instrument was determined during the comparison measurement of newly purchased instruments of the same type. Firstly, the theodolite of the reference instrument should feature high resolution (such as the MINGEO model) and ensure smooth operation of all mechanical components. Secondly, the reference instruments should have relatively high repeatability accuracy among these comparison instruments in these comparison measurements (as repeatability accuracy may also vary among different instruments of the same model). Finally, the baseline obtained from the reference instrument is relatively stable and has the smallest difference compared to the baseline of all instruments in the same batch. Based on the above considerations, the instrument with the best overall performance was chosen as the reference. After being selected as a reference instrument, it has been used as the standard instrument for GNC in all comparison works. And in comparison, skilled observers with proficient techniques were employed, and observer replacements were minimized to reduce the impact of operator errors.

l151 additional azimuth marker alignment > uncertainties resulting from the additional azimuth marker alignment and theodolite levelling

**Response:** This sentence has been corrected.

l163 12 years > 12 inter-comparison campaigns [or similar, the comparisons cover 15 years from 2010-2024]

**Responds:** Thank you for your suggestion. The period from 2010 to 2024 is 15 years, but for some reasons, no comparison was conducted in 2011, 2013, and 2021, resulting in only 12 years of data. The expression you proposed is more accurate, and we have made modifications in the text. The revised sentence is "This study evaluates 12 comparisons data covering 15 years from 2010 to 2024 (no comparison was conducted in 2011, 2013, and 2021)."

Table 3 Define instrument classes (codes).

**Responds:** The instrument classes are no longer used in the revised manuscript, and the parameters found in the manual are directly used for calculation, as shown in Table 2. The specific explanation is as follows: "The classification code 'DJ' in Table 3 (now Table 2) represents the theodolite used for geodetic surveying, derived from the first letters of the two Chinese words 'Dadiceliang'(geodetic surveying) and 'Jingweiyi' (theodolite), with numbers representing the maximum permissible standard deviation for one measurement cycle. For example, 'DJ1' represents a theodolite with a maximum permissible standard deviation of 1" for one measurement cycle."

l178 Type B standard uncertainty: What are Type A and B standard uncertainties?

**Responds:** Additional explanations have been provided in the text regarding Type A and B standard uncertainties. The supplementary content is as follows:
"Type A uncertainty is a type of uncertainty evaluated through statistical methods (e.g., standard deviation of repeated measurement data) to assess the reliability and dispersion of measurement results. Its evaluation relies on the statistical analysis of repeated experimental data. While Type B uncertainty is based on non-statistical methods (e.g., instrument calibration certificates, empirical formulas, or known error limits), often combined with prior information or professional judgment."

l180 What is Delat in Eq. (6)? Provide a reference.

**Responds:** The $\Delta$ in Eq.(6) (now Eq. (8)) is the maximum permissible standard deviation of the theodolite within one measurement cycle. The $\Delta$ in Eq.(6) (now Eq. (8)) has been replaced with $\delta$ in the revised manuscript to distinguishing from the with $\Delta$ which representing instrument difference in the following text. The reference is also provided.

l184 Provide a reference.

**Responds:** The reference has been provided.

l187 There are further instrumental uncertainties, e.g. magnetic contamination of the theodolite body.

**Responds:** Some descriptions about magnetic contamination have been added in the reviewed manuscript. "In addition, there may be other uncertainties that affect the observations, such as magnetic contamination of the theodolite body. Although these affects cannot be quantitatively estimated, they will also be reflected in the measurement results. Therefore, the maximum permissible standard deviation will be temporarily used to estimate the uncertainty of the theodolite intrinsic errors."

Eqs. (10) and (11): x is not defined.

**Responds:** The definition of x in Eqs.(10) and (11) (now Eqs. (12) and (13)) has been supplied. In Eq.(10) (now Eqs. (12)), $x$ is the baseline value calculated according to Eq. (4) and corrected for pillar difference, $N$ is the number of baseline values for each instrument. instruments involved in the comparison. In Eq.(11) (now Eqs. (13)), $\overline{x}$ is the average baseline value of each instrument, $N$ is the number of instruments involved in the comparison.

l198 station specific > pillar specific?

**Responds:** The pillar specific is right.

l217 operator induced > ,the operator induced
**Response:** The word has been corrected.

l256 Orange and green dots: there are no dots in the figure!
**Responds:** This sentence has been revised to "The light orange and light green filled areas represent the distribution of differences for D and I, respectively."

l258 additional azimuth marker alignment: and levelling
**Responds:** This difference arises from the additional azimuth marker alignment step, and the accuracy of the vertical circle setting (at 90° or 270°) required for declination measurements, which introduces greater operator variability.

Figure 5 Are there any trends in the human errors? Difficult to see.
**Responds:** There is no significant trend change in personnel operation errors. In order to facilitate checking whether there is any change in its trend, the drawing type of Fig. 5 (now fig. 8) has been changed.

---

## Editor Decision (ED1)

**Comments on "Evaluating data quality and reference instrument robustness: insights from 12 years DI magnetometer comparisons in the Geomagnetic Network of China" by Y. He *et al.***

The authors have improved the manuscript in response to the points raised by the two reviewers. I find the contents of the revision themselves are sufficient. As the reviewer noted, what makes this paper highly unique is their valuable dataset acquired over a long period at numerous observatories across China. Potentially providing useful information to the geomagnetic observation community, the paper itself is considered suitable for publication.

To make the paper publishable eventually, please address the following two points. Given the scale of the required changes, I impose them as a major revision. While this will place additional burden on the authors, I expect they will make a positive effort to achieve publication.

First, it is unclear what the term "reference instrument" which is the subject of this paper and included in the title, refers to. It first appears in line 51 of Section 1, but it is only a "reference instrument" in a general sense, and can even be read as referring to a proton magnetometer. Section 1 should state that DIM No.0 (the "standard reference instrument" mentioned in line 171) is evaluated as the example for this study.

Table 1 shows "GNC" as the observation station code for DIM No.0. It is unclear at which specific station's reference pillar the measurements were made. Does this mean that the tested DIMs were brought to a specific central station for comparative observations? If so, please clearly highlight the central station in Figure 2. Since pillar differences across remote observatories are generally unobtainable, I presume that the tested DIMs were gathered at one location each year for the intercomparison. If the inter-station DIM comparison were made remotely, clearly describe so in the subsection 2.4, as well as in the abstract.

Second, on top of the original inadequacy of the paper's structure, and the substantial additions made during revising process greatly increased its length, making the author's arguments very difficult to follow (actually, there are so many "Additionally" and "Furthermore" throughout). The presentation should be refined by addressing the following points:

- Rewrite the abstract which is currently seriously disorganized. Just give the outline of paper concisely. No specific figures are necessary. No outlook is necessary. Get straight to the reference instrument robustness. The text of the conclusion section rather has the taste of an abstract. Is the period 12 years or 15 years?

- Keep Section 2 dedicated only for the methodology, as its title says. The three subsections 2.1 to 2.3 review the absolute measurement. I don't think they are essential for discussing the author's findings. They could be eliminated, or compressed to form a subsection in Appendix.

- Let Subsection 2.5 be independent from Section 2 for the methodology. It can be presented in a Results section.

- The authors could move the details of Sebsections 3.1 and 3.2 to Appendix, describing the specific formulations for the uncertainty estimation as an application of the ISO guideline for the current analysis. In the main text, the procedure can be introduced simply, referring to Appendix. It would be the subsection 3.3 that the authors would like to highlight the most and draw reader's attention to.

The above are my suggestions, which are not the condition for publication. The authors do not have to take all of them as they are. In those cases, nevertheless, replies with reasonable explanations are expected.

**Minor issues**

**Line 246:** A new paragraph would start with "This study..."

**Line 288:** Incomplete sentence.

**Line 288:** Misspelling "conduced".

**Table 2:** Misspelling "Resulution".

**Line 467:** The authors' first names and last names are reversed.

---

## Author Response (AR2)

**Responds to the comments**

Thank you to associate editor for your careful review. The original text was revised and marked it in red font. The following is a point-to-point response to the comments.

**Comments on "Evaluating data quality and reference instrument robustness: insights from 12 years DI magnetometer comparisons in the Geomagnetic Network of China" by Y. He et al.**
The authors have improved the manuscript in response to the points raised by the two reviewers. I find the contents of the revision themselves are sufficient. As the reviewer noted, what makes this paper highly unique is their valuable dataset acquired over a long period at numerous observatories across China. Potentially providing useful information to the geomagnetic observation community, the paper itself is considered suitable for publication.

To make the paper publishable eventually, please address the following two points. Given the scale of the required changes, I impose them as a major revision. While this will place additional burden on the authors, I expect they will make a positive effort to achieve publication.

First, it is unclear what the term "reference instrument" which is the subject of this paper and included in the title, refers to. It first appears in line 51 of Section 1, but it is only a "reference instrument" in a general sense, and can even be read as referring to a proton magnetometer. Section 1 should state that DIM No.0 (the "standard reference instrument" mentioned in line 171) is evaluated as the example for this study.
**Response:** In order to avoid ambiguity in understanding, regarding the term "reference instrument", which originally appeared in Section 1, the author added a new explanation (now line 44-46) and defined it as DIMs, and unified the terminology throughout the text. At the same time, relevant descriptions were added for No.0 in Table 1(now line 91-92).

Table 1 shows "GNC" as the observation station code for DIM No.0. It is unclear at which specific station's reference pillar the measurements were made. Does this mean that the tested DIMs were brought to a specific central station for comparative observations? If so, please clearly highlight the central station in Figure 2. Since pillar differences across remote observatories are generally unobtainable, I presume that the tested DIMs were gathered at one location each year for the intercomparison. If the inter-station DIM comparison were made remotely, clearly describe so in the subsection 2.4, as well as in the abstract.
**Response:** In Table 1, GNC is not a station code; it merely indicates that the standard instrument belongs to GNC. The authors have added relevant explanations in lines 91-92 of the text ("The No. 0 in Table 1 is the reference instrument, and the code represents GNC rather than the observatory."). The instrument comparison requires bringing all instruments to a specific station for comparing, rather than conducting it remotely. A related description has been added in lines 85-86 of the text ("During each comparison, all fluxgate theodolites instruments are brought to a specific observatory (code QIX) with excellent observation environment and compared with the reference instrument designated by the GNC."), and it is also highlighted in Figure 2 (now Figure 1).

Second, on top of the original inadequacy of the paper's structure, and the substantial additions made during revising process greatly increased its length, making the author's arguments very difficult to follow (actually, there are so many "Additionally" and "Furthermore" throughout). The presentation should be refined by addressing the following points:

- Rewrite the abstract which is currently seriously disorganized. Just give the outline of paper oncisely. No specific figures are necessary. No outlook is necessary. Get straight to the reference instrument robustness. The text of the conclusion section rather has the taste of an abstract. Is the period 12 years or 15 years?

**Response:** The author has rewritten the abstract and conclusion section according to the editor's suggestions. The comparison work lasted for 15 years, with a total of 12 comparisons. Due to certain reasons, no comparison was organized in 3 years. To avoid unclear expression, rephrase the original sentence in the abstract (line 10) and provide detailed explanations in lines 102-102 of the text.

- Keep Section 2 dedicated only for the methodology, as its title says. The three subsections 2.1 to 2.3 review the absolute measurement. I don't think they are essential for discussing the author's findings. They could be eliminated, or compressed to form a subsection in Appendix.
- Let Subsection 2.5 be independent from Section 2 for the methodology. It can be presented in a Results section.

**Response:** Following the editor's advices, only the textual descriptions of the measurement and comparison methods have been retained in Section 2. Most of the content from the original subsections 2.2, 2.3, and 2.4, particularly the calculation procedures, has been moved to Appendix A. The original subsection 2.5 has been restructured as an independent new section (now Section 3), which is solely used to present the statistical analysis results.

- The authors could move the details of Sebsections 3.1 and 3.2 to Appendix, describing the specific formulations for the uncertainty estimation as an application of the ISO guideline for the current analysis. In the main text, the procedure can be introduced simply, referring to Appendix. It would be the subsection 3.3 that the authors would like to highlight the most and draw reader's attention to.

**Response:** The specific operational steps and calculation processes related to the application of uncertainty analysis methods from the original subsections 3.1 and 3.2 have been moved to Appendix B. And the original subsections 3.3 has been restructured as an independent new section (now Section 4), which primarily focuses on presenting the key results obtained from the uncertainty assessment methodology.

The above are my suggestions, which are not the condition for publication. The authors do not have to take all of them as they are. In those cases, nevertheless, replies with reasonable explanations are expected.

**Response:** Thank you to the associate editor for providing constructive suggestions on the structure of our manuscript. The authors fully agree with these advices and believe that the revised content of the article will be clearer and more readable.

**Minor issues**
Line 246: A new paragraph would start with "This study..."

Line 288: Incomplete sentence.

Line 288: Misspelling "conduced".

Table 2: Misspelling "Resulution".

Line 467: The authors' first names and last names are reversed.

**Responds:** All the errors mentioned above have been corrected in the revised manuscript.

---

## Author Response (AR3)

**Responds to the comments**

Thank you to associate editor for your careful review again. The following is a point-to-point response to the comments. The original text was revised and marked it in red font.

The manuscript has been considerably improved after the revision work this time. Before publishing I request just a few minor modifications:
- L10: "An analysis was conducted on..." to be changed to "Statical analyses were conducted in sequence on..."

**Response:** Based on this recommendation, the author made revisions in the text.
- Table1: It is confusing to give "GNC" in the "Observatory Code" column for the reference instrument. Leaving it blank or with a horizontal line makes it clearer that No.0 is special.

**Response:** In Table 1, the "GNC" in the "Observatory Code" column for the reference instrument has been removed and replaced with a long dash.